# Assessment of Ki-67 Proliferative Index in Cytological Samples of Nodal B-Cell Lymphomas

**DOI:** 10.3390/diagnostics14151584

**Published:** 2024-07-23

**Authors:** Mojca Založnik, Simona Miceska, Simon Buček, Nataša Nolde, Mojca Gjidera, Ulrika Klopčič, Zorica Čekić, Živa Pohar Marinšek, Gorana Gašljević, Veronika Kloboves Prevodnik

**Affiliations:** 1Department of Cytopathology, Institute of Oncology, 1000 Ljubljana, Slovenia; 2Faculty of Medicine, University of Ljubljana, 1000 Ljubljana, Slovenia; 3Department of Pathology, Institute of Oncology, 1000 Ljubljana, Sloveniaggasljevic@onko-i.si (G.G.); 4Faculty of Medicine, University of Maribor, 2000 Maribor, Slovenia

**Keywords:** eyeballing, fine-needle aspiration biopsy, flow cytometry, immunocytochemistry, immunohistochemistry, non-Hodgkin nodal B-cell lymphomas, lymph node biopsy, manual counting, Ki-67 proliferative index

## Abstract

Background: The Ki-67 proliferative index (PI) is part of the diagnosis of nodal B-cell lymphoma (nBCL), but its determination in cytological samples is not standardized. We aimed to establish an approach for the accurate determination of the Ki-67 PI in cytological slides to differentiate between indolent and aggressive nBCLs. Methods: Patients diagnosed with nBCL by fine-needle aspiration biopsy and subsequent excision biopsy were included. Cell suspensions were prepared from biopsy samples for CD3/Ki-67 double immunocytochemical staining and flow-cytometric verification of lymphoma B-cell counts. The Ki-67 PI was assessed by manual counting and eyeballing in cytology and eyeballing in histology. The cut-off values for the differentiation between aggressive and indolent lymphomas were determined for each method. Results: A strong correlation between manual and flow-cytometric counting of lymphoma B cells was confirmed (interclass correlation coefficient (IC coef.) = 0.78). The correlation of the Ki-67 PI determined in cytological and histological slides was also strong (IC coef. > 0.80). Histologically, 55 cases were classified as indolent and 31 as aggressive nBCLs. KI-67 PI cut-off values of 28.5%, 27.5%, and 35.5% were established for manual counting and eyeballing in cytology and eyeballing in histology, respectively, with high sensitivity and specificity. Conclusions: The Ki-67 PI, assessed by manual counting and eyeballing in cytological samples, accurately differentiates between indolent and aggressive nBCLs.

## 1. Introduction

The Ki-67 proliferative index (PI) is an important factor for predicting clinical behavior and grading different human malignancies, including non-Hodgkin’s lymphomas [1]. However, its prognostic value may vary across different lymphoma subtypes, and the relationship between Ki-67 expression and outcome with various subtypes of lymphoma are still contradictory and inconclusive [2,3,4,5,6]. For example, in follicular lymphomas (FLs), the 2017 WHO classification recommends the Ki-67 PI as an adjunct to histological grading to enable the recognition of FL cases with low grade morphology and high Ki-67 PI (≥30%) [7]. However, this approach was abandoned by the 5th edition of WHO classification (WHO5) [8]. On the other hand, mantle cell lymphomas (MCLs) are classified as indolent mature B-cell neoplasms, even though there are morphologically more agressive subtypes of MCLs, where a Ki-67 PI > 30% is associated with an adverse prognosis [8,9,10,11,12]. In diffuse large B-cell lymphomas (DLBCLs) the role of Ki-67 PI as a prognostic marker remains controversial [13,14,15].

The Ki-67 proliferative index (Ki-67 PI) is usually assessed immunohistochemically (IHC) in histological samples or immunocytochemically (ICC) in cytological samples. In both methods, the positive reaction is given as a percentage of positive cells’ nuclei among all cells in a defined area and is referred to as the Ki-67 PI. In histology and lymphoma diagnostics, together with morphological assessment, the Ki-67 PI could be used to evaluate the proliferative activity of lymphoma cells and could be valuable for clinical decision making and individual prognostic evaluations [16]. Noteworthily, lymph node fine-needle aspiration biopsy (FNAB) is an essential part of the diagnosis and management of patients with lymphadenopathy, particularly when ancillary techniques complement morphological assessments. It is a well-established method for the differentiation between reactive lymphocytic proliferations and lymphomas to confirm recurrent lymphomas, monitor treatment responses, and determine prognostic and predictive factors [17]. Although the majority of primary nodal B-cell lymphomas (nBCLs) are diagnosed histologically, there are certain clinical scenarios where the FNAB with ancillary methods could be used as a rapid, reliable, and accurate diagnostic modality, especially in lymphoid proliferations in hard-to-reach sites (retroperitoneal lymph nodes, the intraocular region, etc.); fragile, elderly patients with multiple comorbidities; and lymphomas in effusions, such as breast implant-associated anaplastic T-cell lymphomas and primary effusion lymphomas [17].

However, there are limited data on cytological studies related to the classification and grading of non-Hodgkin nBCLs (NH nBCLs) in FNAB samples that include the determination of the Ki-67 PI. Different approaches have been proposed in a limited number of published studies. For instance, the cell size and the percentage of small and large cells were used as cytomorphological criteria to differentiate between small-cell indolent and large-cell aggressive NH nBCLs [18]. Other authors used the Ki-67 PI for the classification of NH nBCLs and as a prognostic indicator [19]. Although a correlation between Ki-67 PI evaluation and the grade of NH nBCLs in FNAB samples was found in some studies, mainly for lymphoma subtypes such as DLBCL, FL, MCL, and marginal-zone lymphoma (MZL) [20,21,22,23], the percentage of Ki-67-positive cells showed a substantial overlap between aggressive and indolent lymphomas. Most of the above-mentioned studies are older than 20 years. Since then, the NH nBCL classification has changed substantially, many new lymphoma entities have emerged, and new ancillary methods have been introduced as a part of the routine diagnostics of these lymphomas [8,24]. However, the role of the Ki-67 PI in stratifying HN nBCLs by FNAB into indolent and aggressive lymphomas remains poorly established. We found only one recently published study by Mao et al., who studied the KI-67 PI by multiparametric flow cytometry (FC). They showed a good correlation with histological results and set a cut-off value for the accurate differentiation between indolent and aggressive lymphomas and indolent and transformed lymphomas [25]. However, to our knowledge, there are no new studies on Ki-67 PI determination using the ICC method. Hence, our objective was to establish an ICC-based approach by manual counting and eyeballing for the accurate determination of Ki-67 in cytological slides and to set a Ki-67 PI cut-off value that would help stratify the most frequent subtypes of NH nBCLs into indolent and aggressive ones.

## 2. Materials and Methods

### 2.1. Patients

Patients who were diagnosed with NH nBCL from FNAB lymph node samples with a subsequent excisional biopsy in the period from January 2013 to July 2014 at the Institute of Oncology Ljubljana (IOL) were included in this study. Since the classification of lymphomas has changed in the last ten years [8], when the practical part of this study was performed, histological and cytological (FNAB) diagnoses were reviewed blindly and revised by one experienced hematopathologist (G.G.) and one experienced cytopathologist (V.K.P.), respectively, following ‘The International Consensus Classification of Mature Lymphoid Neoplasms’, ‘The 5th Edition of the WHO Classification of Hematolymphoid Neoplasms’, and the newly published ‘Proposal for the Performance, Classification, and Reporting of Lymph Node Fine-Needle Aspiration Cytopathology: The Sydney System’ [8,17,26]. For this purpose, histological and FNAB slides, as well as FC results, were retrieved from the Pathology and Cytopathology Archives and the patient medical record (PMR) database at the IOL, respectively. According to the revised histological diagnoses, the cases were divided into indolent and aggressive lymphomas. Indolent lymphomas comprise cases of chronic lymphocytic leukemia (CLL), follicular lymphoma (FL) (grade I, II, and IIIa), mantle-cell lymphoma (MCL) (classic variant), and marginal-zone lymphoma (MZL), while aggressive lymphomas comprise cases of diffuse large B-cell lymphoma (DLBCL), grade IIIb FL, MCL (blastoid and pleomorphic variant), and lymphoblastic lymphoma (B-LBL) [8,26].

Each patient signed an informed consent form. This study was approved by the Committee for the Scientific Evaluation of Clinical Research Protocols at the IOL (ERID-KSOPKR/65, OIRIKE00143).

### 2.2. Study Design

Following the excisional biopsy for histological diagnosis, a small part of each excised lymph node was cut out and sent to the Department of Cytopathology at the IOL, where it was disintegrated to prepare a cell suspension for further ICC staining and FC analysis. Dual ICC staining with Ki-67 and CD3 antibodies was performed to assess the percentage of Ki-67-positive lymphoma B cells (Ki-67 PI) and T cells. The ICC staining results were assessed by two different methods: (1) manual counting and (2) eyeballing. Since the accurate distinction between lymphoma B cells and reactive B cells on the ICC slides was not possible, we assumed that all B cells represented lymphoma B cells. The accuracy of manual counting on ICC slides was confirmed by comparing the percentages of lymphoma B cells and T cells determined by the ICC and FC methods. Furthermore, the results of the Ki-67 PI determined by the manual counting and eyeballing of the ICC slides were compared with the Ki-67 PI results assessed by the eyeballing of the IHC slides. The eyeballing was performed semi-quantitatively. Moreover, a cut-off value of the Ki-67 PI to differentiate between the indolent and aggressive NH nBCLs was determined. A visual description of the study design is given in Figure 1.

### 2.3. Flow Cytometry

The excised lymph node samples were disintegrated to prepare cell suspensions using the gentle MACS Dissociator (Miltenyi Biotec, Bergisch Gladbach, Germany), as previously described by our group, and the samples were prepared according to the standard protocol for immunophenotyping cytological samples at the IOL at the time of diagnosis [27,28,29]. An antibody panel consisting of four tubes was used, with each tube containing four different antibodies, all from BD Biosciences. The first tube contained an isotypic control for B cells. The second tube included CD3-FITC (catalog number: 564106), CD19-PE (catalog number: 561802), CD45-PerCP-Cy5.5 (catalog number: 345777), and CD20-APC (catalog number: 340941). The third tube had kappa-FITC (catalog number: 643774), lambda-PE (catalog number: 642925), CD19-PerCP-Cy5.5 (catalog number: 332780), and CD10-APC (catalog number: 332777). The fourth tube consisted of FMC7-FITC (catalog number: 332786), CD23-PE (catalog number: 341008), CD19-PerCP-Cy5.5 (catalog number: 332780), and CD5-APC (catalog number: 340583). Importantly, only the FC results from the second tube were used for the comparison of the count of lymphoma B cells and T cells with the ICC manual counting and eyeballing.

The samples were acquired on 6-color FC BD FACS Canto (BD Biosciences, San Jose, CA, USA) and analyzed with the DIVA software v 8.0 (BD Biosciences, San Jose, CA, USA). If FC measurements were already performed on FNAB samples as part of the routine cytological lymphoma diagnostics (as in most of the cases), the FC results were retrieved from the PMR database. The percentages of B cells and T cells were calculated as the percentages of all lymphocytes in the measured samples.

### 2.4. Immunocytochemistry

An aliquot of each cell suspension was used to prepare three methanol-fixed cytospins for ICC staining. One of the cytospins was used as a negative control, the second for dual Ki-67/CD3 ICC staining, and the last for repeated testing, if required. The mouse monoclonal Ki-67 antibody (clone MIB-1, 1:200, Agilent, Waldbronn, Germany, catalog number: M724001-2; incubation time: 16 min at 37 °C) was used to assess the Ki-PI in lymphoma B cells, and CD3 (LN10, Leica Biosystems, Abingdon, UK, 1:100, catalog number: LE-CD3-565-L-CE; incubation time: 32 min at 37 °C) to assess the percentage of T cells. The Ki-67-positive reaction was accomplished by the brown chromogen 3,3′-diaminobenzidine (DAB) (iView DAB detection kit; catalog number: 760-091; Tucson, AZ, USA) and the CD3-positive reaction by the red chromogen Fast Red (the enhanced alkaline phosphatase red detection kit, catalog number: 760-501, Ventana, Roche Diagnostics, Tucson, AZ, USA). The antigen retrieval was accomplished by Ultra Cell Conditioning 1 (CC1, catalog number: 950-124, Ventana, Roche Diagnostics, Tucson, AZ, USA) at 95 °C for 8 min. ICC staining was performed with a Benchmark GX immunostainer (Ventana, Roche Diagnostics, Tucson, AZ, USA).

### 2.5. Immunohistochemistry

The IHC staining for Ki-67 (clone MIB-1, 1:40, Dako, catalog number: M724001-2; incubation time: 16 min at 37 °C) was performed on 3 μm thick sections with the Ventana Benchmark XT immunostainer (Ventana, Roche Diagnostics, Tucson, AZ, USA) using the ultraView Universal DAB Detection Kit (catalog number: 760-500, Ventana, Roche Diagnostics, Tucson, AZ, USA). The antigen retrieval with CC1 was applied under the same conditions as described for ICC. The histopathological assessment of the Ki-67 PI was given as an average of the Ki-67 expression in lymphoma cells evaluated by eyeballing by one experienced hematopathologist (MG).

### 2.6. Ki-67 PI Scoring in Cytological Samples

In dual Ki-67/CD3 ICC-stained slides, the Ki-67 PIs of NH nBCLs were estimated in two ways: (1) by manual counting and (2) by eyeballing. For the manual counting, 1000 cells were counted altogether. Among the 1000 cells, the number of CD3-positive T cells (red cytoplasmic staining) and the number of lymphoma B cells (with an unstained cell cytoplasm with or without brown/Ki-67 nuclear staining) were determined. The number of CD3-positive cells was determined by counting, while the number of lymphoma B cells was calculated by subtracting the CD3-positive T cells from the total count of 1000 cells. The percentage of Ki-67-positive lymphoma B cells was then determined by dividing the number of Ki-67-positive lymphoma cells by the number of lymphoma B cells and expressed as a percentage. For the eyeballing assessment, the percentage of Ki-67-positive lymphoma B cells was determined semi-quantitatively by two experienced cytopathologists (Z.Č. and U.K.), independently and blindly, neither of whom had information on the histological results. If there was no significant difference between their assessments, only the results from Z.Č. were used.

### 2.7. Statistical Analysis

All statistical analyses were performed using IBM SPSS Statistics, version 25 (SPSS Inc., Chicago, IL, USA). The obtained data were analyzed with classical descriptive methods for nonparametrically distributed data: medians and ranges. The Wilcoxon test was used to compare the Ki-67 PIs of aggressive and indolent NH nBCLs. The intraclass correlation coefficient (IC coef.) was used to calculate the agreement between the continuous variable Ki-67 PI determined by counting, eyeballing, and histology and the agreement between the counting of lymphoma B cells and FC. The IC coef. was described as very weak (0.00–0.19), weak (0.20–0.39), moderate (0.40–0.59), strong (0.60–0.79), and very strong (0.80–1.0). The ROC curve with the criterion closest to 0.1 and a *Youden index = (sensitivity + specificity) − 1* were used to determine the optimal cut-off value for the Ki-67 PI between indolent and aggressive lymphomas [30]. A Youden index above 50% indicated that the determined cut-off value met the empirical benchmarks for being administered for diagnostic purposes.

## 3. Results

### 3.1. Patients

Altogether, 86 patients with FNAB and histology of the same lymph node were included in this study, 43 of which were male and 43 were female. The median age of the patients at the time of diagnosis was 61 (range: 8–80) years. The histological diagnoses are shown in Table 1. In total, 55 cases were stratified as indolent (29 FLs (low grade), 10 CLLs, 5 MCLs (classical variant), and 11 MZLs) and 31 cases as aggressive (27 DLBCLs, 1 B-LBL, 1 FL (GIIIb), and 2 MCLs (blastoid variant)) lymphomas.

The revised cytopathological examination accurately distinguished lymphomas from reactive non-lymphoma cases in 85/86 (99%) of cases. One case was diagnosed as suspicious for NH nBCL but was histologically confirmed as marginal-zone lymphoma (MZL).

Moreover, in 76/86 (88%) of cases, the lymphoma type was correctly classified by revised cytopathological examination. All mantle-cell lymphomas (MCLs), as well as chronic lymphocytic leukemias (CLLs), were classified correctly (100%). Follicular lymphomas (FLs) and DLBCLs were classified with high accuracy via cytology in 29/30 (97%) and 22/27 (81%) of cases, respectively. The highest discordance between cytology and histology was obtained for the diagnosis of MZLs, where only 6/11 (64%) of cases were classified correctly. One MZL case in cytology was diagnosed as suspicious for NH nBCL and three cases as FLs because of CD10 positivity determined by FC. Regarding the course of the disease, 25/31 (80%) of cases were cytologically correctly classified as aggressive and 55/55 (100%) as indolent. The correlations between the cytopathological and histological diagnoses are shown in Table 1.

### 3.2. Correlation between the Percentages of Lymphoma B Cells Determined by Manual Counting and Flow Cytometry

Our results showed a median of 77.7% (range: 37.3–100%) of lymphoma B cells, determined by manual counting, and 67.0 (range: 14.1–94.2%) when determined by FC, which resulted in a strong correlation between the results of both methods (IC coef. = 0.78). For T cells, we observed a value of 22.3% (range: 0–62.0%) with manual counting and 33.1% (range: 4.0–99.0%) with FC, confirming a very strong correlation between the methods (IC coef. = 0.83). A detailed description of the results is given in Table 2.

### 3.3. Correlation between the Ki-67 PIs Determined in Cytological and Histological Samples

The medians and ranges of the Ki-67 PIs in NH nBCLs determined in the cytological samples by manual counting and eyeballing and in the histological samples by eyeballing are shown in Table 3. The Ki-67 PIs significantly differed between indolent and aggressive NH nBCLs when determined by all three methods (*p* < 0.0001). Specifically, there was a very strong correlation between the Ki-67 PIs in NH nBCLs determined by manual counting and eyeballing in cytological samples (IC coef. = 0.915). A very strong correlation was also observed when comparing the Ki-67 PIs determined by eyeballing in cytological samples and eyeballing in histological samples (IC coef. = 0.809), as well as between manual counting in cytological samples and eyeballing in histological samples (IC coef. = 0.811) (Table 3). The dual Ki-67/CD3 ICC staining is shown in Figure 2.

### 3.4. Determination of Ki-67 Cut-Off Value to Distinguish between Indolent and Aggressive Non-Hodgkin Nodal BCLs by ROC Curve Analysis

The cut-off values gained with the ROC curve analysis for the Ki-67 PI cut-off to distinguish aggressive from indolent NH nBCLs for cytological manual counting and eyeballing and histological eyeballing were 28.5%, 27.5%, and 35.0%, respectively (Figure 3; Table 4). The values for the sensitivity, specificity, and the Youden index were all above 90% for both manual counting and eyeballing in cytology. Specifically, for manual counting, the sensitivity, specificity, and Youden index were 100%, 90.9%, and 90.9%, respectively. For eyeballing, these values were 96.2%, 94.5%, and 90.7%, respectively. For histology, the sensitivity, specificity, and Youden index were 100%, 85.5%, and 85.5%, respectively (Table 4). Using the Ki-67 cut-off value of 28.5, all aggressive cases (31/31) and 53/55 (96%) of indolent cases were correctly reclassified by manual counting. When considering the Ki-67 cut-off value of 27.5% in eyeballing, 30/31 of aggressive cases (97%) and 53/55 of indolent cases (96%) were correctly reclassified. In histology, our cut-off of 35.5% correctly classified all aggressive cases but not all indolent cases.

It is important to emphasize that the six aggressive NH nBCLs (one FL G III and five DLBCLs), which were initially misclassified as FLs or MZLs and considered indolent by the revised cytological diagnosis, were correctly reclassified as aggressive types by applying both a Ki-67 PI cut-off value of 28.5% for manual counting and 27.5% for eyeballing.

However, there were also three cases where the correct indolent lymphoma types were confirmed by revised cytological diagnosis, but their Ki-67 PI values were above the cut-offs, leading to their misclassification as aggressive types (Table 5, bottom bold-framed part of the table).

## 4. Discussion

The diagnostic evaluation of NH nBCLs is primarily based on histological examination, with the determination of the Ki-67 PI being an important component of this histological examination [1,16,31]. The Ki-67 PI in NH nBCLs, as well as in other neoplasms, can be assessed in different ways, mainly by manual counting, which involves determining the exact number of cells expressing Ki-67 in a specified area, but also by eyeballing, which is a semi-quantitative method that provides a rough estimation based on visual assessment alone, without using precise measuring [32,33]. Both assessment methods rely on internationally established guidelines. For instance, for breast carcinoma, the manual counting of the Ki-67 PI has been standardized as a part of the diagnostic procedures, where 2 × 500 tumor cells in IHC-stained sections are considered the »gold standard« for the prediction of prognosis [34,35]. Similarly, the Ki-67 PI predicts outcomes in patients diagnosed with advanced-stage MCLs who are treated with anti-CD20 immunochemotherapy [9,36]. Eyeballing in histology is also commonly used since it is a less time-consuming approach for frequent routine practice.

However, in some clinical scenarios, such as recurrent lymphomas, the presence of lymphoid proliferations in hard-to-reach sites (retroperitoneal lymph nodes, the intraocular region, etc.) and fragile, elderly patients with multiple comorbidities, FNAB could be used as a rapid, reliable, and accurate diagnostic method, especially when complementary techniques supplement morphological assessments [37]. Wang et al. performed a meta-analysis on the correlation of NH nBCL diagnoses between cytology and histology. Their analysis showed that the combination of FNAB with complementary methods resulted in a high agreement between most frequent cytological and histological diagnoses [37], but not many publications can be found on this matter. Our analysis showed that cytological examination was highly accurate in categorizing the most frequent NH nBCLs (99%), as well as in correctly categorizing the lymphoma type when classifying MCLs and CLLs (100%), but was discordant when classifying FLs (19%), DLBCLs (3%), and, especially, MZLs (46%). Moreover, when FNAB diagnoses were used for the classification of NH nBCLs in indolent and aggressive lymphomas, all 55 indolent lymphomas were correctly classified. However, 6/31 of aggressive lymphomas were misclassified. Therefore, the Ki-67 PI may improve classifications between indolent and aggressive lymphomas.

ICC manual counting and eyeballing are complementary methods to IHC for the determination of the KI-67 PI, but unlike histology, cytology lacks comprehensive data and standardization for these two methods, despite the frequent need for cytologists to determine if NH nBCLs are indolent or aggressive. Drawing a parallel with existing histological guidelines, our study aimed to evaluate the manual counting of the Ki-67 PI in 1000 cells and the eyeballing of Ki-67 PI determination in cytospins and to compare the cytological results with those established via histological eyeballing, aiming to establish a quick and simple method for routine practice.

Our comparison of the Ki-67 PI results via cytology versus histology allowed us to correctly classify lymphoma grades, which is especially crucial for aggressive lymphoma cases. Manual counting and eyeballing in cytology resulted in a very high correlation with each other (IC coef. > 0.9) and also with eyeballing in histology (IC coef. > 0.8), indicating that both cytological assessment methods are equally reliable and suitable for the routine determination of the Ki-67 PI in cytological samples and in discordant scenarios; as such, the Ki-67 PI may be helpful to correctly establish cytological diagnoses.

Furthermore, our results support published data on cytospins as an appropriate technique to assess Ki-67 PIs [19,38]. Recently, our group published data on the good agreement of Ki-67 assessments in cytospins and cytoblocks in cytological (effusion) samples and histological (tumor tissue) samples in serous high-grade carcinomas [39]. These data support the potential of cytological samples as a promising source to analyze the Ki-67 PI of malignant cells in addition to conventional cytomorphological and immunophenotypic assessments. Thus, useful information for planning the therapeutic approach could be obtained based on the cytospin-determined Ki-67 PI.

Meanwhile, there have been several attempts to demonstrate the usefulness of the Ki-67 PI in differentiating indolent from aggressive lymphomas using eyeballing in histology as well as cytology. So far, Broyde et al. [40] evaluated the Ki-67 PI in 319 newly diagnosed cases of NH nBCLs in histological sections and showed a statistically significant increase in the mean Ki-67 PI values from 26.6% for indolent lymphomas to 67.2% for aggressive lymphomas and to 97.6% for very aggressive lymphomas. They established a Ki-67 PI of 45% to distinguish indolent from aggressive lymphomas [40]. The cut-off determined by our analysis was somewhat lower (35%), but remained with a 100% sensitivity and 85.5% specificity, correctly classifying all aggressive lymphomas.

In principle, a higher Ki-67 PI is expected in aggressive lymphomas compared with indolent lymphomas in FNAB samples. However, there is no defined Ki-67 PI cut-off value in cytology that allows a reliable distinction between these two categories. Mihaljević et al. reported an increasing trend in the Ki-67 PI in FNAB samples from lymphomas consisting of small cells to those composed of small cells with notched nuclei to those composed of large cells with histopathological equivalents corresponding to aggressive lymphomas [38]. Skoog et al. [19] found a good correlation between the Ki-67 PI and the cytological classification of indolent and aggressive lymphomas. However, within all lymphoma subgroups, there was a clear variation in the proportion of Ki-67-positive cells, leading to some overlap between indolent and aggressive lymphomas. Furthermore, the Ki-67 PI in reactive lymphadenitis varied between 1 and 50%, and it was concluded that the Ki-67 PI alone cannot be used to differentiate between benign and neoplastic proliferation. Abdulah et al. reported a MIB-1 PI ranging from 1% and 99%, with a mean value of 31.5% [41]. The range was 1–50% for indolent lymphomas, 20–99% for aggressive lymphomas, and 70–99% for highly aggressive lymphomas. We established Ki-67 PI cut-off values of 28.5 and 27.5% by manual counting and eyeballing, respectively, that generally correctly classified aggressive lymphomas and discrepancy in ≤five indolent lymphomas. In cases with discordant cytomorphological assessments, these cut-off values can improve diagnosis by accurately determining the disease grade and guiding therapeutic decisions. When the cytomorphology and Ki-67 PI are both in favor of indolent and aggressive lymphoma, there is a strong probability that the grading of the lymphoma is correct. However, in cases where the cytomorphology and Ki-67 PI give discrepant results, clinical features such as lactate dehydrogenase (LDH) levels and positron emission tomography–computed tomography (PET-CT) results may help in distinguishing between indolent and aggressive lymphomas [42]. In uncertain cases, a biopsy is recommended.

The good cytological–histological correlation of the Ki-67 PI results in NH nBCLs may be due to the thick aggregates of lymphoma cells, which are easier to analyze than lumpy carcinoma cells. However, discrepancies do occur. We speculate that common reasons for discrepant Ki-67 PI results might include MZLs with a higher percentage of large, transformed cells; transformation from indolent into aggressive lymphomas; and the partial infiltration of lymph nodes by lymphomas, causing sampling issues. Additionally, other NH nBCL subtypes, such as angioimmunoblastic lymphomas, may contain many reactive cells, leading to decreased Ki-67-positive cell levels. Despite the low incidence of these lymphomas, the Ki-67 PI remains a valid marker of the tumor cell growth fraction in most NH nBCL entities.

Meanwhile, the prognostic value of the Ki-67 PI is crucial for various lymphoma subtypes [2,3,4,5,6]. For instance, in FLs, the 2017 WHO classification recommends using the Ki-67 PI as an adjunct to histological grading to identify cases with a low-grade morphology but a high (≥30%) Ki-67 PI [7]. MCLs, typically classified as indolent mature B-cell neoplasms, can include morphologically aggressive blastoid and pleomorphic variants, where a Ki-67 PI of >30% indicates an adverse prognosis [8,9,10,11]. Therefore, Ki-67 PI determination is essential in certain lymphoma entities, and our study demonstrates its equal accuracy in cytology and histology for these assessments.

However, despite our promising findings, our study had limitations. Both manual counting and eyeballing rely on human visual assessment, which can be prone to occasional inaccuracies. Additionally, improper sample visualization due to factors like cell aggregation or debris further complicates analysis. Other evaluation methods, such as FC and digital image analysis (DIA), remain promising but are still facing challenges. For instance, there is a publication by Mao et al. [25] demonstrating the utility of multicolor FC in Ki-67 PI assessment cytological samples. They showed a strong agreement between FC-determined Ki-67 PI values in cytology and those in tissue samples analyzed by IHC. Their method effectively distinguished between indolent and aggressive lymphoma types and identified transformations in indolent lymphomas at lower cut-off values (21.25% for type differentiation and 7.65% for transformation) compared with the cut-offs determined in our analysis. We speculate that a possible reason for the higher cut-off values in our study could be that with ICC, we could not exclude the count of some reactive, non-neoplastic B cells in the samples. FC, as a more sophisticated method, allows gating strategies to exclude reactive B cells in a sample, but most cytological laboratories have limited availability of flow cytometers and specialized analysts. The ICC approach and Mao’s FC method demonstrated high accuracy in distinguishing indolent from aggressive lymphomas. Each cytological laboratory can select the method that best suits its equipment and resources. Meanwhile, DIA publications on Ki67 show promise for improving accuracy and consistency, since excellent agreement with pathology has been shown, especially for breast cancer [43,44,45]. But despite its promise, DIA faces challenges, such as technical complexity and cost, limited access for most laboratories, a lack of standardized evaluation methods and software, the need for specialized training, calibration and validation issues, sensitivity to artifacts, and data management difficulties, delaying its routine clinical practice implementation for a few more years [46]. Thus, using ICC to determine Ki-67 in cytological samples is still a valuable technique, providing practical insights into the disease aggressiveness and transformation potential.

## 5. Conclusions

Our findings indicate that the cytological assessment of the Ki-67 PI, using both manual counting and eyeballing, is equally as reliable as histology for diagnosing NH nBCLs in cytological samples. It effectively distinguishes between indolent and aggressive lymphomas with our proposed cut-offs of 28.5% and 27.5%, respectively. These methods could significantly enhance the accuracy of cytomorphological diagnosis, particularly in aggressive NH nBCL cases. However, further multicentric studies are essential to investigate larger patient cohorts across various lymphoma subtypes and standardize cytological Ki-67 PI cut-off values.

## Figures and Tables

**Figure 1 diagnostics-14-01584-f001:**
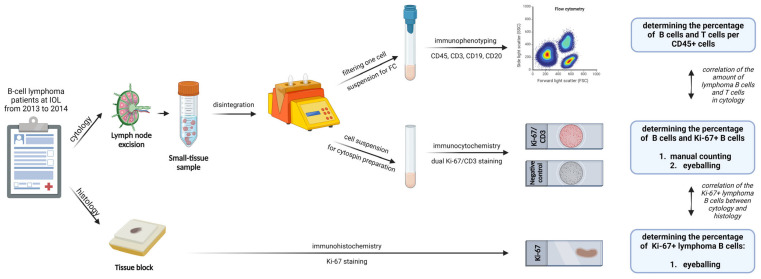
Visual description of the study design. Created with Biorender.com.

**Figure 2 diagnostics-14-01584-f002:**
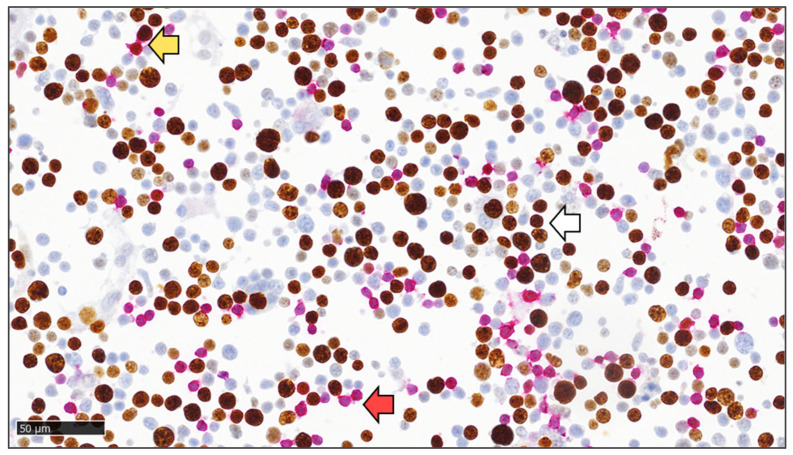
Dual Ki-67/CD3 immunocytochemical staining on methanol-fixed cytospins of cytological samples. T cells can be identified by unstained nuclei and red-stained cytoplasm for CD3 (as indicated by the red arrow), or by brown-stained nuclei for Ki-67 and red-stained cytoplasm for CD3 (as indicated by the yellow arrow). Lymphoma B cells can be identified as cells with brown-stained nuclei for Ki-67 and unstained cytoplasm (as indicated by the white arrow) (400× magnification).

**Figure 3 diagnostics-14-01584-f003:**
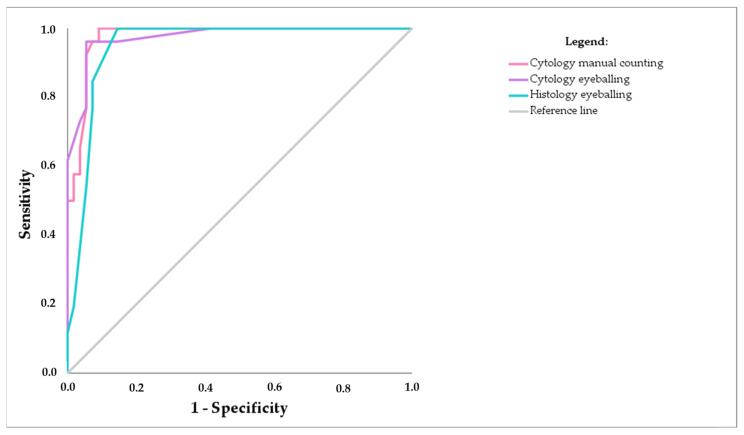
ROC curves for the differentiation between indolent and aggressive nodal BCHLs using Ki-67 PI for manual counting and eyeballing in cytological samples.

**Table 1 diagnostics-14-01584-t001:** Correlation between cytological and definitive histological diagnoses of all NH nBCLs included in this study.

	Histological Diagnosis									
Cytological diagnosis	**Final Diagnosis**	**DLBCL**	**B-LBL**	**FL G IIIb**	**FL-LG**	**CLL**	**MCL bl.**	**MCL cl.**	**MZL**	**Total**
	DLCBL	22	-	-	-	-	-	-	-	22
	B-LBL	-	1		-	-	-	-	-	1
	FL	1	-	1	29	-	-	-	3	34
	CLL	-	-	-	-	10	-	-	-	10
	MCL	-	-	-	-	-	2	5	-	7
	MZL	4	-	-		-	-	-	7	11
	Suspicious for BCL	-	-	-	-	-	-	-	1	1
	Total	27	1	1	29	10	2	5	11	86

Abbreviations: DLBCL—diffuse large B-cell lymphoma; B-LBL—lymphoblastic lymphoma; FL G IIIb—follicular lymphoma (grade IIIb); FL-LG—follicular lymphoma (low grade); CLL—chronic lymphocytic leukemia; MCL bl.—mantle-cell lymphoma (blastoid); MCL cl.—MCL (classical); MZL—marginal-zone lymphoma; BCL—B-cell lymphoma; N—number of cases.

**Table 2 diagnostics-14-01584-t002:** Percentages of lymphoma B cells and T cells determined by manual counting and flow cytometry.

Method	Manual Counting	Flow Cytometry	Manual Counting vs. Flow Cytometry Correlation	
**Cells**	Median (Range), (%)	Median (Range), (%)	IC Coef.	95% CI; *p*-Value
**Lymphoma B cells (*N* = 75)**	77.7 (37.3–100)	67.0 (14.1–94.2)	0.780	6.297 (0.662–0.925); *p* > 0.001
**T cells (*N* = 75)**	22.3 (0–62.7)	33.1 (4.0–99.0)	0.830	1.711 (−0.245–0.726); *p* = 0.081

Abbreviations: %—percentage; CI—confidence interval; IC coef.—interclass correlation coefficient.

**Table 3 diagnostics-14-01584-t003:** Medians and ranges of Ki-67 PIs for all NH nBCLs included in this study and for indolent and aggressive lymphomas.

Sample Type	Method	Ki-67 PI in NH nBCLs (Median Percentage (Range))			Intraclass Coefficients Between the Methods		
		All Cases (%)	Indolent Nodal BCL Cases Only (%)	Aggressive Nodal BCL Cases Only (%)			
Cytological	Manual counting	20.0 (3.0–98.0)	11.0 (3.0–46.0)	58.5 (31.0–98.0)	0.915 *		0.811 ***
	Eyeballing	20.0 (1.0–98.0)	10.0 (1.0–45.0)	60.0 (20.0–98.0)		0.809 **	
Histological	Eyeballing	30.0 (2.0–100)	15.0 (2.0–80.0)	70.0 (40.0–100)			

Abbreviations: BCL—B-cell lymphoma; Ki-67 PI—Ki-67 proliferative index; *—intraclass coefficient between cytology manual counting and cytology eyeballing; **—intraclass coefficient between cytology eyeballing and histology eyeballing; ***—intraclass correlation between cytology manual counting and histology eyeballing.

**Table 4 diagnostics-14-01584-t004:** Ki-67 PI cut-off values for the differentiation between indolent and aggressive NH nBCLs determined by ROC curve analysis in cytological samples.

Method	Cut-Off Value for Ki-67 PI	AUC	95% CI Interval; *p*-Value	Sensitivity (%)	Specificity (%)	Youden Index *	Aggressive Cases, Discordant with the Determined Cut-Off	Indolent Cases, Discordant with the Determined Cut-Off
Manual counting	23.0	0.976	0.949–1.000; *p* < 0.000	100	85.7	85.5	0	8
	25.0			100	87.3	87.3	0	7
	**28.5 ***			**100**	**90.9**	**90.9**	**0**	**3**
	32.5			96.2	90.0	87.1	1	5
	35.3			96.2	91.7	88.9	1	3
Cytological eyeballing	17.5	0.975	0.945–1.000; *p* < 0.000	100	58.2	58.2	0	25
	22.5			96.2	85.5	81.6	1	9
	**27.5 ***			**96.2**	**94.5**	**90.7**	**1**	**3**
	35.0			80.8	94.5	75.3	5	3
	45.0			76.9	94.5	71.5	6	3
Histological eyeballing	22.5	0.950	0.903–0.997; *p* < 0.000	100	69.1	69.1	0	19
	27.5			100	70.9	70.9	0	18
	**35.0 ***			**100**	**85.5**	**85.5**	**0**	**8**
	42.5			92.3	89.1	81.4	3	6
	47.5			88.5	90.9	79.4	4	5

Abbreviations: CI—confidence interval; Ki-67 PI—Ki-67 proliferative index; *—the bold cut-off values for Ki-67 PI are the chosen cut-offs for each method, determined by ROC curve analysis, specificity, sensitivity, and Youden index values.

**Table 5 diagnostics-14-01584-t005:** Comparison of the Ki-67 PIs between cytological manual counting, cytological eyeballing, and histological eyeballing for the 10 NH nBCLs cases with discrepant subtypes of final diagnoses between cytology and histology (upper bold-framed part of the table) and for cases with correct indolent cytological diagnoses but discrepant KI-67 PI results after applying the cut-offs (bottom bold-framed part of the table).

	Diagnosis	Cytology			Histology		
	Case Number	Diagnosis	Ki-67 PI by Manual Counting (%)	Ki-67 PI by Eyeballing (%)	Diagnosis	KI-67 PI by Eyeballing (%) and Grade Classification	
Discordant cytological diagnoses	54	FL	50	60	FL G IIIb	60	Aggressive
	9	MZL	43	60	DLBCL	40	Aggressive
	16	MZL	31	30	DLBCL	40	Aggressive
	31	MZL	42	50	DLBCL	70	Aggressive
	37	FL	39	30	DLBCL	70	Aggressive
	46	MZL	53	40	DLBCL	80	Aggressive
Discordant KI-67 PI results compared with cytological diagnoses	61	FL	**37**	10	FL	20	Indolent
	72	MZL	**46**	**40**	MZL	25	Indolent
	79	FL	26	**45**	FL	40	Indolent

Abbreviations: DLBCL—diffuse large B-cell lymphoma; FL—follicular lymphoma; FL G IIIb—follicular lymphoma (grade IIIb); Ki-67 PI—Ki-67 proliferative index; MZL—marginal-zone lymphoma; BCL—B-cell lymphoma. Note: Discordant KI-67 PI results of the correctly classified indolent cytological diagnoses are in bold.

## Data Availability

The data will be made available after considering the aim of further use.

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
