# Peer review of "Assessment of Ki-67 Proliferative Index in Cytological Samples of Nodal B-Cell Lymphomas"

_diagnostics, 2024, doi:10.3390/diagnostics14151584_

Round 1
Reviewer 1 Report
Comments and Suggestions for Authors
The study is interesting, well performed and is sound in the exposition of the results and discussion. The manuscript may be accepted provided of minor changes.
Minor points:
The study has been performed on cytological suspensions prepared from surgical biopsies where the authors assume that the whole procedure may be effective on cytological fine-needle aspiration samples, this point should be further discussed.
A former study by Sun et al., performed using Ki67 on fine-needle samples of follicular lymphoma (FL), distinguished FL grade 1 from FL grades 2 and 3 whereas it did not clearly distinguish between grades 2 and 3. This study should be considered in the introduction and in the discussion. (Sun W et al. Grading follicular lymphoma on fine needle aspiration specimens. Comparison with proliferative index by DNA image analysis and Ki-67 labeling index. Acta Cytol. 2004;48(2):119-26).
Flow cytometry was used to assess the phenotype of corresponding lymphoma entities and to assess the CD3 positive T-cells but did play a direct role in the Ki67 assessment. This point should be clarified.
Comments on the Quality of English Languagenone
Author Response
Reviewer 1
Open Review
(x) I would not like to sign my review report
( ) I would like to sign my review report
Quality of English Language
( ) I am not qualified to assess the quality of English in this paper
( ) English very difficult to understand/incomprehensible
( ) Extensive editing of English language required
( ) Moderate editing of English language required
(x) Minor editing of English language required
( ) English language fine. No issues detected
|
Yes |
Can be improved |
Must be improved |
Not applicable |
|
|
Does the introduction provide sufficient background and include all relevant references? |
(x) |
( ) |
( ) |
( ) |
|
Is the research design appropriate? |
(x) |
( ) |
( ) |
( ) |
|
Are the methods adequately described? |
(x) |
( ) |
( ) |
( ) |
|
Are the results clearly presented? |
(x) |
( ) |
( ) |
( ) |
|
Are the conclusions supported by the results? |
( ) |
(x) |
( ) |
( ) |
Comments and Suggestions for Authors
The study is interesting, well performed and is sound in the exposition of the results and discussion. The manuscript may be accepted provided of minor changes.
Minor points:
The study has been performed on cytological suspensions prepared from surgical biopsies where the authors assume that the whole procedure may be effective on cytological fine-needle aspiration samples, this point should be further discussed.
Indeed, our study was conducted on cytological suspensions prepared from surgical biopsies to ensure that both cytological and histological Ki-67 PI scores would be assessed on the same lymph node samples, since histological assessment was used as a reference. However, our extensive experience with fine-needle aspiration biopsy (FNAB) samples at the Department of Cytopathology at the Institute of Oncology Ljubljana has demonstrated that they are just as reliable as lymph node biopsy samples (unpublished data). This is because both types of biopsies involve random sampling, making the sampling error equally possible in both cases. Of course, it would be very interesting to analyze and compare the exact correlation between FNAB and lymph node surgical biopsies' Ki-67 PI, and publish these findings. For now, we have included this consideration as a limitation in our study, and added it in the Discussion section.
A former study by Sun et al., performed using Ki67 on fine-needle samples of follicular lymphoma (FL), distinguished FL grade 1 from FL grades 2 and 3 whereas it did not clearly distinguish between grades 2 and 3. This study should be considered in the introduction and in the discussion. (Sun W et al. Grading follicular lymphoma on fine needle aspiration specimens. Comparison with proliferative index by DNA image analysis and Ki-67 labeling index. Acta Cytol. 2004;48(2):119-26).
Thank you for the suggestion. We mentioned this publication in as suggested (reference number 23). In Sun et al.’s study [23], which was performed 2004, they considered comparison of FL grade 1, 2 and 3, by showing that with digital image analysis of propidium iodide stained nuclei for determination of cell fraction in S phase of cell cycle, and ICC determination of KI-67 PI, FL grade 1 can be clearly distinguished from grades 2 and 3, but this was not as clear for distinguishing between grades 2 and 3. However, in our study FL were grouped differently, according to the newest WHO5 classification, which regarding FL aggressiveness states the follows:
‘FL grade I, grade II and IIIa were considered as low grade FL, and only FL IIIb was considered as aggressive, since in clinical practice is treated as DLBCL. Here we show citation from the WHO5 classification: Traditionally, FL has been graded as FLI, FLII, FLIIIA, and FLIIIB, according to the quantification of absolute numbers of centroblasts / transformed cells in 10 consecutive high-power fields (HPFs). However, there is accumulating evidence of the lack of reproducibility in counting centroblasts, and thus of grading itself. Many studies have indicated no statistically significant difference in clinical outcomes between FL1, FL2, and FL3A, which are treated similarly in modern clinical trials. Furthermore, there is compelling evidence that FLIIIa is biologically related to FLI and FLII, with similar immunohistochemical and genetic profiles and frequent coexistence in the same affected lymph node. At present, there is no definitive evidence to support the distinction between FLI, FLII, and FLIIIa, and hence to mandate grading. FLI, FLII and FLIIIa together constitute the FL subtype, which is defined by a mixture of centrocytes and centroblasts in various proportions, but in which centrocytes must be unequivocally present.'
Flow cytometry was used to assess the phenotype of corresponding lymphoma entities and to assess the CD3 positive T-cells but did play a direct role in the Ki67 assessment. This point should be clarified.
We assessed the immunophenotype of lymphoma cells in FNAB samples of the lymph nodes (which were sent for biopsy and histological verification) to make preoperative lymphoma diagnosis by our antibody panel for immunophenotyping, which consisted of four tubes, with each containing four different antibodies (detailed in the section 2.3. Flow cytometry, please follow track changes):
- Tube 1: isotypic control for B cells,
- Tube 2: CD3-FITC, CD19-PE, CD45-PerCP-Cy5.5, and CD20-APC,
- Tube 3: kappa-FITC, lambda-PE, CD19-PerCP-Cy5.5 and CD10-APC,
- Tube 4: FMC7-FITC, CD23-PE, CD19-PerCP-Cy5.5 and CD5-APC.
Importantly, only the FC results from the second tube were used for the comparison of the count of lymphoma B-cells and T cells with the ICC manual counting and eyeballing.
The Ki-67 assessment was performed on suspension biopsy from the lymph-node samples. The Ki-67 PI was determined on lymphoma B-cells only. CD3 staining was used to help us differentiate lymphoma B-cells from reactivate CD3 positive lymphocytes, therefore KI-67 PI of CD3+ T-cells did not affect the KI-67 PI determined on lymphoma B-cells.
PS. We would like to add that we observed a typographical error in the count of indolent and aggressive lymphomas in the manuscript. The numbers were previously listed as 56 and 30, respectively, but have now been corrected to 55 and 31. It is important to clarify that this correction does not affect the results or conclusions of the manuscript.
Comments on the Quality of English Language: none
Submission date: 01 July 2024
Date of this review: 06 Jul 2024 16:55:17
Reviewer 2 Report
Comments and Suggestions for Authors
This study analyzed the proliferation index expression by ki67 in B-cell lymphomas, and found that the quantification of ki67 could differentiate between indolent from aggressive variants. Of note, it is expected that aggressive variants have higher proliferation index. The manuscript is well written, it is easy to read and to understand. To improve the text, the authors could address the following comments:
(1) Line 33. Regarding “The Ki-67 proliferative index (PI) is an important factor for predicting clinical behavior and grading different human malignancies, including non-Hodgkin's lymphomas 34 (NHLs) [1]”. Please confirm this statement with other references. Ki67 prognosis is related closely with mantle cell lymphoma. But in other lymphomas, not so sure. Please revise and cite the corresponding publications.
(2) Line 38. Regarding “Ki-67 PI”. Sorry to ask, but what is “PI”? Proliferation index?
(3) Regarding “FNAB with ancillary methods could be used as a rapid, reliable, and accurate diagnostic modality”. While FNAB can be useful at the beginning of the diagnosis, there is a loss of histological features. Because of the large number of lymphoma subtypes, using only FNAB could be considered a “diagnostic risk” (according to the opinion of some pathologists).
(4) Line 74. If you cite the proposal classification of WHO5, that is based on the WHO4 revised, you may also have to cite the ICC2022. https://doi.org/10.1182/blood.2022015851
(5) Lines 77 to 85. Regarding the diagnosis of the cases, was it a combination of FNA, biopsy, and flow cytometry in all cases?
(6) Line 103. Please describe “eyeballing” technique of quantification. Is it s semi-quantitative assessment by two people?
(7) In section 2.3. What was the FC primary antibody for ki67 evaluation?
(8) Please add catalog numbers of the different reagents that were use in Material and Methods.
(9) Line 195. In FL-LG, low grade. Is grade IIIa included in “low grade”?
(10) Section 3.2. As I understand, ki67 was also evaluated in reactive T-cells? What is the benefit/intention to evaluate ki67 in T-cells (on a B-cell lymphoma study)?
(11) Line 305. Regarding “Our analysis showed that FNAB cytological examination is highly accurate in diagnosing nodal BCLs”. Please note the there are many other types of mature B and T-cell neoplasms. For example, Hodgkin lymphoma is not included in this series.
(12) In the diagnosis of the different lymphomas subtypes, did you use immunophenotype characterization with other markers? For example, for MCL, did you stain for cyclind1?
(13) In the abstract, please add the number of cases of indolent and aggressive variants of b-cell lymphoma.
Author Response
Reviewer 2
Open Review
(x) I would not like to sign my review report
( ) I would like to sign my review report
Quality of English Language
( ) I am not qualified to assess the quality of English in this paper
( ) English very difficult to understand/incomprehensible
( ) Extensive editing of English language required
( ) Moderate editing of English language required
( ) Minor editing of English language required
(x) English language fine. No issues detected
|
Yes |
Can be improved |
Must be improved |
Not applicable |
|
|
Does the introduction provide sufficient background and include all relevant references? |
( ) |
(x) |
( ) |
( ) |
|
Is the research design appropriate? |
(x) |
( ) |
( ) |
( ) |
|
Are the methods adequately described? |
(x) |
( ) |
( ) |
( ) |
|
Are the results clearly presented? |
( ) |
(x) |
( ) |
( ) |
|
Are the conclusions supported by the results? |
(x) |
( ) |
( ) |
( ) |
Comments and Suggestions for Authors
This study analyzed the proliferation index expression by ki67 in B-cell lymphomas, and found that the quantification of ki67 could differentiate between indolent from aggressive variants. Of note, it is expected that aggressive variants have higher proliferation index. The manuscript is well written, it is easy to read and to understand. To improve the text, the authors could address the following comments:
- Line 33. Regarding “The Ki-67 proliferative index (PI) is an important factor for predicting clinical behavior and grading different human malignancies, including non-Hodgkin's lymphomas 34 (NHLs) [1]”. Please confirm this statement with other references. Ki67 prognosis is related closely with mantle cell lymphoma. But in other lymphomas, not so sure. Please revise and cite the corresponding publications.
We revised and cited new references as suggested.
(2) Line 38. Regarding “Ki-67 PI”. Sorry to ask, but what is “PI”? Proliferation index?
Indeed, PI stands for proliferation index, the abbreviation is first mentioned and explained in the introduction part, line 33. From then on, we use the abbreviation through whole manuscript.
(3) Regarding “FNAB with ancillary methods could be used as a rapid, reliable, and accurate diagnostic modality”. While FNAB can be useful at the beginning of the diagnosis, there is a loss of histological features. Because of the large number of lymphoma subtypes, using only FNAB could be considered a “diagnostic risk” (according to the opinion of some pathologists).
We revised the introduction section (please follow the track changes) and changed the primary statement, which previously claimed that "cytology is an accurate diagnostic modality for lymphomas." We appreciate your insightful feedback and are grateful for the opportunity to improve this misstatement.
(4) Line 74. If you cite the proposal classification of WHO5, that is based on the WHO4 revised, you may also have to cite the ICC2022. https://doi.org/10.1182/blood.2022015851
Thank you for the remark, we have added the additional citation as suggested.
(5) Lines 77 to 85. Regarding the diagnosis of the cases, was it a combination of FNA, biopsy, and flow cytometry in all cases?
Indeed, yes. In all cases included in the study, FNAB and flow cytometry were performed preoperatively to make the lymphoma diagnosis. In fact, our flow-cytometric panel consisted of four tubes, with each containing four different antibodies (detailed in the section 2.3. Flow cytometry):
- Tube 1: isotypic control for B cells,
- Tube 2: CD3-FITC, CD19-PE, CD45-PerCP-Cy5.5, and CD20-APC,
- Tube 3: kappa-FITC, lambda-PE, CD19-PerCP-Cy5.5 and CD10-APC,
- Tube 4: FMC7-FITC, CD23-PE, CD19-PerCP-Cy5.5 and CD5-APC;
Only the FC results from the second tube were used for the comparison of the count of lymphoma B-cells and T cells with the ICC manual counting and eyeballing. Here our goal was not to assess Ki-67 PI with flow cytometry but just to assess the accuracy of ICC count itself. One of the reasons we did not choose to additionally estimate Ki-67 using flow cytometry is that it was not a routine practice in our lab, and we had previously faced challenges in implementing it. To avoid confusion for other readers, we have improved the explanation in the study design.
Moreover, in all cases where mantle cell lymphoma was suspected, cyclin D1 was additionally determined by ICC on methanol-fixed cytospins. We did not mention this in the manuscript because we believed it was beyond our primary scope. If you consider it necessary to include a more detailed explanation on this matter, please let us know.
In addition to FNAB, lymph node biopsy and histology were also performed for each case.
(6) Line 103. Please describe “eyeballing” technique of quantification. Is it a semi-quantitative assessment by two people?
Eyeballing is explained in detail in section 2.6. Ki-67 PI scoring in cytology samples: ‘For the eyeballing assessment, the percentage of Ki-67 positive lymphoma B-cells was determined semi-quantitatively by two experienced cytopathologists (ZC, UK) independently and blindly, neither of whom had information on the histologic results’. Since there were no significant differences between the results of the two cytopathologists, the results of ZC were further used for statistical analysis.
(7) In section 2.3. What was the FC primary antibody for ki67 evaluation?
As mentioned earlier in question number 5, we did not evaluate Ki67 using flow cytometry, our reference for comparison was histology eyeballing. The purpose of using flow cytometry in our study was just to assess the percentage of lymphoma B cells with a method that is quantitative in order to confirm the reliability of determining percentage of lymphoma B cells with manual counting on ICC slides.
(8) Please add catalog numbers of the different reagents that were use in Material and Methods.
Thank you for the observation, we have added the catalog numbers (please follow track changes in the manuscript).
(9) Line 195. In FL-LG, low grade. Is grade IIIa included in “low grade”?
Yes. According to the newest WHO classification which we followed, FL grade I, grade II and IIIa are considered as low grade FL, and only FL IIIb is considered as aggressive, since in clinical practice is treated as DLBCL, as we have described in the section 2.1., "Patients ":
Indolent lymphomas comprise cases of chronic lymphocytic leukemia (CLL), follicular lymphoma (FL) grade I, II and IIIa, mantle cell lymphoma (MCL) classic variant, and marginal zone lymphoma (MZL), while aggressive lymphomas comprise cases of diffuse large B-cell lymphoma DLBCL, FL grade IIIb, MCL blastoid and pleomorphic variant and lymphoblastic lymphoma (B-LBL).
Moreover, we copy citation from the WHO5 classification:
Traditionally, FL has been graded as FL1, FL2, FL3A, and FL3B, according to the quantification of absolute numbers of centroblasts / transformed cells in 10 consecutive high-power fields (HPFs). However, there is accumulating evidence of the lack of reproducibility in counting centroblasts, and thus of grading itself. Many studies have indicated no statistically significant difference in clinical outcomes between FL1, FL2, and FL3A, which are treated similarly in modern clinical trials. Furthermore, there is compelling evidence that FL3A is biologically related to FL1/2, with similar immunohistochemical and genetic profiles and frequent coexistence in the same affected lymph node. At present, there is no definitive evidence to support the distinction between FL1, FL2, and FL3A, and hence to mandate grading. FL1/2 and FL3A together constitute the FL subtype, which is defined by a mixture of centrocytes and centroblasts in various proportions, but in which centrocytes must be unequivocally present.
(10) Section 3.2. As I understand, ki67 was also evaluated in reactive T-cells? What is the benefit/intention to evaluate ki67 in T-cells (on a B-cell lymphoma study)?
As described in section 2.6, "Ki-67 PI Scoring in Cytology Samples," we performed double immunocytochemical staining. Ki-67 positive cells were stained brown. Since B-cells and T-cells are morphologically similar, we stained T-cells red using the anti-CD3 antibody. This allowed us to identify and exclude double-positive cells (Ki-67 positive T-cells), so only single-stained Ki-67 cells were considered lymphoma B-cells. We used this approach because CD3 is a universal marker for T cells, while identifying B cells requires at least two markers (CD19 and CD20), which is impractical with immunocytochemistry. For manual counting, we first counted all brown, Ki-67 positive cells, then counted all double-stained brown and red cells (Ki-67+CD3+ T cells). By subtracting the double-stained cells from the total Ki-67 positive cells, we determined the number of Ki-67 positive lymphoma B cells.
In section 3.2, "Correlation between the Percentages of Lymphoma B-Cells Determined by Manual Counting and by Flow Cytometry," we compared the percentages of lymphoma B cells assessed by manual counting and eyeballing with their percentages determined by flow cytometry. This comparison did not include Ki-67 assessment, but only the percentage of lymphoma B cells and T cells assessed by immunocytochemistry (manual counting and eyeballing) and by flow cytometry, since here we only aimed to evaluate the reliability of the counting of B-cell and T-cells on ICC slides, as flow cytometry is a precise quantitative technique.
(11) Line 305. Regarding “Our analysis showed that FNAB cytological examination is highly accurate in diagnosing nodal BCLs”. Please note the there are many other types of mature B and T-cell neoplasms. For example, Hodgkin lymphoma is not included in this series.
We analyzed only the most frequent types of non-Hodgkin nodal B-cell lymphomas, including FL, CLL, MZL, MCL, and DLBCL. Hodgkin lymphomas were excluded from our study because Ki-67 is not prognostically significant for Hodgkin lymphoma and is not part of the routine diagnostic immunocytochemistry.
(12) In the diagnosis of the different lymphomas subtypes, did you use immunophenotype characterization with other markers? For example, for MCL, did you stain for cyclin D1?
At the time when the study was performed, we used an antibody panel for cytological B-cell lymphoma determination, consisting of four tubes, each containing four different antibodies (see the answer to question 5 and follow the track changes in section 2.3, Flow Cytometry). Nowadays, we have improved this panel to a single tube containing 10 different antibodies, as we have upgraded from a 6-color FACS Canto BD to a 10-color Canto10 BD flow cytometer.
However, for MCL, diagnostically we always perform cyclin D1 staining by immunocytochemistry because our practical experience showed flow cytometry is not the best approach for cyclin D1 assessment. However, we did not include these results in the manuscript because they were beyond the scope of our objectives.
(13) In the abstract, please add the number of cases of indolent and aggressive variants of b-cell lymphoma.
We have added the number of indolent and aggressive cases as requested (please follow track changes in the manuscript).
PS. We would like to add that we observed a typographical error in the count of indolent and aggressive lymphomas in the manuscript. The numbers were previously listed as 56 and 30, respectively, but have now been corrected to 55 and 31. It is important to clarify that this correction does not affect the results or conclusions of the manuscript.
Submission date: 01 July 2024
Date of this review: 08 Jul 2024 18:29:23